# Northern Richness, Southern Dead End—Origin and Dispersal Events of *Pseudolycoriella* (Sciaridae, Diptera) between New Zealand’s Main Islands

**DOI:** 10.3390/insects14060548

**Published:** 2023-06-12

**Authors:** Arne Köhler, Thomas Schmitt

**Affiliations:** 1Senckenberg German Entomological Institute, 15374 Müncheberg, Germany; thomas.schmitt@senckenberg.de; 2Developmental Biology, Institute of Biology, Martin Luther University of Halle-Wittenberg, 06120 Halle (Saale), Germany; 3Entomology and Biogeography, Institute of Biochemistry and Biology, Faculty of Science, University of Potsdam, 14476 Potsdam, Germany

**Keywords:** Australia, colonisation, intraspecific distribution, island biogeography, phylogeography

## Abstract

**Simple Summary:**

The New Zealand species of a genus of black fungus gnats show clear phylogeographic patterns, at the species level and above. North Island harbours more species than South Island, and according to our phylogeographic analyses, was more often the starting point of dispersal events to South Island than vice versa. We therefore deduce that North Island is a radiation centre. Initial colonisations of New Zealand took place three times, most likely starting from Australia, with the earliest in the late Miocene.

**Abstract:**

Sciaridae (Diptera) is a widespread insect family of which some species can reach high abundances in arboreal habitats. This trait, together with their (passive) mobility, enables them to quickly colonise suitable habitats. To reveal the biogeographic history of the New Zealand members of the sciarid genus *Pseudolycoriella*, we analysed three molecular markers of selected species and populations in a Bayesian approach. At the intra- and interspecific levels, we detected a pattern of northern richness vs. southern purity, which has probably developed as a result of Pleistocene glacial cycles. Since the late Miocene, we identified 13 dispersal events across the sea strait separating New Zealand’s main islands. As nine of these dispersal events were south-directed, North Island can be considered the centre of radiation for this genus. An unequivocal re-colonisation of North Island was only observed once. Based on the inclusion of three undescribed species from Tasmania and on previously published data, three colonisations of New Zealand are likely, all of them assumed to be of Australian origin. One of these most probably took place during the late Miocene, and the other two during the late Pliocene or at the Pliocene–Pleistocene boundary.

## 1. Introduction

New Zealand and its biogeography have been investigated by generations of natural scientists. In this context, the origin of its unique biota as well as their distributions have been a focus of interest, and several studies have already addressed the biogeography of New Zealand’s insects (see [1,2,3]). The cause of this uniqueness is New Zealand’s outstanding geological genesis, and in particular its long and ongoing isolation from neighbouring landmasses. At the end of the Mesozoic, Zealandia—a landmass that comprised what today is New Caledonia and New Zealand—separated from Australia and then was further fragmented [1,4]. These fragments then experienced even more drastic geological events. At the Oligocene–Miocene boundary, the future New Zealand was affected by major inundations and almost entirely disappeared in the ocean, although the archipelago was never completely submerged [5,6]. The re-emerging landmasses were strongly affected by plate tectonics, as a new plate boundary had formed in the Eocene, with today’s North Island and the western part of South Island located on the Australian plate, while the remaining part of today’s South Island is located on the Pacific plate. The land masses that represent the two extant main islands were separated by sea most of the time [7,8]. In the Pleistocene, ca. 1 Ma ago [8], eustatic lowering of the sea level formed land connections between the islands for the first time. However, due to the glacial cycles with their associated sea level oscillations, land bridges were ephemeral. Another still persisting consequence of these Pleistocene cycles is the well-known “beech gap”—a region in central South Island where the iconic southern beeches (*Nothofagus* spp.) have not re-colonised glacially devastated areas [9].

In arboreal habitats, Sciaridae are rich in species and in abundance [10]. These globally distributed gnats are minute to medium-sized, and due to their uniform habitus, they are easy to recognize as a group, but determination to the species level usually proves to be challenging. Therefore, it is not surprising that, despite their prominence and commonness, only the small minority of pest species has attracted wider attention. This is regrettable, because due to their ecological role as decomposers [11,12], their ability to establish populations at high abundances [10], and their ability to fly or disperse passively, they seem to be well-suited as model organisms for the assessment of biogeographic events. For instance, airborne dispersal was recorded at altitudes up to 3350 m above the South Pacific [13], and they have been found in arthropod fallout on snow fields near the summit of Mt. Teide on Tenerife [14]. Even the drifting of different developmental stages with plant material, or pure surface-floating, is easily conceivable [15].

A sciarid taxon well-suited for unravelling the colonisation and biogeographic history of New Zealand might be the genus *Pseudolycoriella*. In the course of a recent revision by Köhler [16], the number of known species on these islands rose from seven to thirty-eight. Two of these still do not have valid names, because formal descriptions are lacking: males of these—a prerequisite for species descriptions in sciarids—are not available. Only one of these New Zealand species, *P*. *cavatica*, is also known from other countries. Its wide distribution (recorded from Australia, Hawaii, New Caledonia, Seychelles, South Africa, and Tristan da Cunha [16]), intraspecific molecular data [17], and lack of a closer phylogenetic relationship to any other *Pseudolycoriella* species occurring in New Zealand [16] support the hypothesis that *P*. *cavatica* is an introduced species. Accordingly, this species was omitted from the subsequent analyses. Of the 37 autochthonous and endemic New Zealand species (both described and undescribed), 25 are known from more than one sample site. These species can be roughly assigned to four broad distribution patterns (Table 1). Only three native species are distributed throughout New Zealand. Another four species are found on North Island and the northern part of South Island. Twelve species are restricted to North Island, and six to South Island. The relative distribution of endemics in *Pseudolycoriella* on the two main islands differs significantly from the ratio expected according to Trewick et al. [18], who reviewed the regional insect endemicity of five major regions, but also from the expectation in relation to the area ratio of the two islands (Chi-square goodness of fit test: Χ^2^ ≈ 4.5 resp. Χ^2^ ≈ 4.1, α = 0.05, df = 1). Thus, compared to South Island, North Island has a higher number of species than expected. Even the number of species currently known from only one sample site tends to show this biased distribution across both islands (six species from North Island vs. six from South Island).

Consequently, we decided to attempt a phylogenetic study in order to make these inhomogeneous distribution patterns comprehensible. We address the question of whether there is evidence that these patterns are the result of Pleistocene glaciation, or even of older geological phenomena. We also intend to draw conclusions on New Zealand’s black gnat colonisation history. These data are compared against the background of the already known biogeographic patterns of New Zealand insects.

## 2. Materials and Methods

Specimens were either collected by Catrin and Mathias Jaschhof in 2001 and 2002 (collecting permit #9900/143/3/04 issued by the Department of Conservation, New Zealand) or by Peter A. Maddison from 2014 to 2016. Thus, our biogeographic analyses are based on the material previously used by Köhler [16] for a taxonomic revision and description of new species. In addition to the material from New Zealand, specimens from Tasmania collected with Malaise traps at the Warra Long-term Ecological Research Site [19] were incorporated in the analyses.

DNA extraction and sequencing were performed as in Köhler [16]. We analysed three genes: cytochrome c oxidase subunit I (COI; 658 bp), the domains IV and V of the mitochondrial large subunit rRNA gene (16S; 538 bp), and the nuclear 28S ribosomal gene (28S; 1857 bp). The gene selection was based on the first phylogenetic study of sciarids by Shin et al. [11]. This mixture of rapidly evolving mitochondrial genes and a more conservative nuclear gene seems to be appropriate for the time scale under investigation, as already applied by Köhler [16]. In some cases, residual DNA from the DNA extraction by Köhler [16] was re-sequenced to eliminate ambiguities in COI or 16S sequence of some specimens. For this purpose, the primer Psl_COI_for (5′–ATTATAATTTTTTTYATAGTDATACC–3′) was designed and successfully applied. All sequences are available from GenBank, with the corresponding GenBank accession numbers listed in the electronic supplement (Appendix A).

To enable intraspecific spatial analyses, the software Popart 1.7 [20] was used to generate median-joining haplotype networks [21] based on COI and 16S sequences. For the interspecific relationships and especially for the divergence time estimates, a Bayesian analysis was performed using the multilocus sequence data package *BEAST from the program BEAST 2.6.0 [22]. As two different monophyletic *Pseudolycoriella* species clades exist in New Zealand [16], we conducted two independent *BEAST analyses. Input files were generated with Beauti [23] using the following parameter settings: the HKY model as a substitution model, estimated base frequencies, four gamma categories, a strict clock model, and a birth–death model for the speciation process. If specimens of a species had identical haplotypes in all three genes, only the haplotype combination of one specimen was included in the input files. The input files comprise sequences from 95 specimens of 35 species (35 specimens of 13 species belonging to the *P*. *macrotegmenta* clade, and 60 specimens of 22 species of the second monophyletic group).

The (current) lack of fossil sciarids from New Zealand (this taxon is not mentioned by Schmidt et al. [24]) and of *Pseudolycoriella* globally does not permit direct calibration. Thus, fixed substitution rates were used: 0.01345 for the mitochondrial markers and 0.0006 for the nuclear marker. These rates were derived by Papadopoulou et al. [25] for the same genes used in the present study based on the biogeography of tenebrionoid beetles driven by the formation of the Mid-Aegean trench 9–12 Ma ago. The Markov Chain Monte Carlo (MCMC) was set to a length of 20,000,000 generations with a sampling interval of 7500 iterations. The obtained samples were checked for sufficient effective sample size (ESS) values using the software Tracer 1.7.1 [26]. The initial 10% of the sampled generations were discarded as burn-in. The coalescence tree was generated with TreeAnnotator [23].

To draw inferences about the ancestral origin of New Zealand *Pseudolycoriella* species, the RASP 4.2 software [27,28] was used, applying the statistical dispersal–extinction–cladogenesis model (S-DEC). For this purpose, all specimen records were assigned to four major regions, i.e., Tasmania, North Island, the northern part of South Island, and the southern part of South Island including Stewart Island. Thereby, South Island was divided along 43°S, roughly corresponding to the northern edge of the “beech gap” [9]. A contiguous distribution is assumed for *P*. *subtilitegementa*, which was recorded in southern North Island and southern South Island [16].

For easier comprehension, all nodes of the two resulting chronograms are numbered consecutively, and branches are named after their delimiting nodes (indicated in the text with braces).

## 3. Results

### 3.1. Intraspecific Biogeographic Patterns

Four species were well-suited for the analyses of intraspecific biogeographic patterns. Two of these are distributed across both main islands, i.e., *P*. *tonnoiri* and *P*. *zealandica*. For these, 26 COI and 20 16S sequences as well as 44 COI and 42 16S sequences, respectively, were available. The calculated haplotype networks revealed a subdivision into populations on both sides of Cook Strait (Figure 1). While *P*. *tonnoiri* possesses two haplotypes, which only differ in one single substitution in COI (Figure 1A), *P*. *zealandica* was found to have a tripartite population structure with clear spatial structure (Figure 1B). The genetic diversity of the latter declines from North Island to South Island: five different haplotypes of both COI and 16S were recorded for North Island from two geographically very close localities (the corresponding points overlap in Figure 1B), while only three COI haplotypes and one 16S haplotype were obtained for South Island.

The two species *P*. *tewaipounamu* and *P*. *sudhausi* solely inhabit South Island and exhibit remarkable haplotype patterns. *Pseudolycoriella tewaipounamu* (50 COI, 46 16S sequences) has three genetic lineages exhibiting a clear phylogeographic pattern: in the north-west (Buller District; yellow circles in Figure 2A), the south-west (Westland and Southland Districts; red circles), and the south-east (Clutha District; blue circles) of South Island. The genetic diversity of the populations within these lineages varies strongly. Specimens from the north-western lineage exhibit the greatest differences and diversity in COI; the south-eastern lineage is also genetically diverse and well differentiated from the north-western lineage, while the sequences of the south-western lineage are all identical and represent a haplotype shared with the north-western lineage (Figure 2A). The 16S haplotype network indicates a pattern consistent with that of COI, although the differences at the population level are less marked or absent. The only major difference is the occurrence of a second haplotype in the south-western linage, resulting from an additional transition.

For *P*. *sudhausi* (19 COI, 16 16S sequences), three clusters are distinguished: on Stewart Island (blue circles in Figure 3A), and in the south-western (south of Fiordland; red circles) and western parts of South Island (north of Fiordland and on the West Coast; yellow circles). The shortest known geographic distance between sampling localities belonging to different genetic clusters lies in Fiordland and is only about 23 km (Figure 3B). Specimens of a southern lineage population were sampled along the Eglinton River West Branch valley, while members of the northern lineage were caught in the Hollyford River valley.

### 3.2. Interspecific Differentiation and Phylogeny

The first monophylum of autochthonous New Zealand *Pseudolycoriella* species comprises the species closely related to *P*. *macrotegmenta* and is partitioned into three subclades (Figure 4). One includes the majority of nine species (*P*. *macrotegmenta* s. str. clade) and stands in a sister relationship with a subclade composed of three still—undescribed species from Tasmania. The most basal subclade is represented by a single New Zealand species (*P*. *jaschhofi*) and originated in a species split during the Pliocene or Pleistocene (node 1, 2.98 Ma; 95% HPD interval: 1.87–4.30 Ma). Of the subsequent eleven speciation events, three most likely took place in the early Pleistocene, while the others occurred not earlier than one million years ago. According to a RASP analysis, the most recent common ancestor (MRCA) of the *P*. *macrotegmenta* s. str. clade most likely inhabited North Island (node 5, 63.1%). The MRCA of the Tasmanian species and the *P*. *macrotegmenta* s. str. clade was probably distributed across Tasmania and North Island (node 2, 43.5%). The second most likely scenario—a solely Tasmanian distribution—has a probability value of 17.9%. For node 1, i.e., the most basal one, the probability values for the different distribution scenarios are close together and do not exceed 12.5%; thus, the distribution at this node remains unsolved.

The chronogram of the second monophyletic group of New Zealand *Pseudolycoriella* species (Figure 5) replicated the tripartite structure already shown in Köhler [16]. Accordingly, the naming of the three subclades after species whose names have been known for decades is retained (*P*. *zealandica*, *P*. *bispina*, and *P*. *jejuna* clade; indicated by different colours in Figure 5). Compared with the *P*. *macrotegmenta* clade, this monophyletic group has a longer speciation history. The first speciation occurred in the late Miocene (node 18, 9.95 Ma; 95% HPD interval: 7.45–12.38 Ma), and was followed by 20 species splits until the Pleistocene, evenly distributed across time. The phylogeographic analysis yielded a probability value of 51.8% that the MRCA of this group inhabited North Island (node 18). The *P*. *zealandica* clade is the youngest of these three taxa; the basal species split took place in the early Pliocene (node 19, 4.95 Ma; 95% HPD interval: 3.64–6.35 Ma). Its MRCA probably inhabited entire New Zealand (72.8%). The basal splits of the other species clades were assigned to the late Miocene: the speciation of the MRCA of the *P*. *bispina* clade was estimated at 7.90 Ma (node 24; 95% HPD interval: 6.02–9.62 Ma) and that of the *P*. *jejuna* clade at 6.09 Ma (node 33; 95% HPD interval: 4.92–7.41 Ma). The most likely distribution of the MRCA of the *P*. *bispina* clade was North Island (87.3%). The probability values of the distribution of the *P*. *jejuna* clade are as follows: entire New Zealand 41.9%; North Island and the northern part of South Island 22.8%; and only North Island 29.0%. Based only on the highest probability values assigned to each MRCA of the 20 species splits that occurred in the three *Pseudolycoriella* clades, nine MRCAs were distributed in North Island, nine in both main islands, and two (nodes 38 and 41) in South Island.

Appendix A give the time estimates and probability values for each region of origin for all species distributions shown in Figure 4 and Figure 5.

## 4. Discussion

### 4.1. Ice Ages and Their Influence on New Zealand’s Scarids

The two analysed species, which are distributed across both large islands, i.e., *P*. *tonnoiri* and *P*. *zealandica*, both show spatial structuring of their haplotypes, with a clear split along Cook Strait. However, the level of diversity of haplotypes and their geographic distribution differs greatly between these species. In *P*. *tonnoiri*, the lineages differ in only a single base substitution. However, this species belongs to the *P*. *macrotegmenta* s. str. clade, whose species are generally not strongly genetically differentiated. This led Köhler [16] to the conclusion that their radiation is more recent than in other New Zealand *Pseudolycoriella* species groups. Consistent with this, the split between the two lineages of *P*. *tonnoiri* has an estimated median age of 28 ka, making it the most recent in our analysis (node 13 in Figure 4). As its ancestor (node 9 in Figure 4 and Figure 6A–C) most likely inhabited only North Island, the most parsimonious explanation is that this region was the origin of *P*. *tonnoiri*. Interestingly, *P*. *tonnoiri* was recently reported from Auckland Island, more than 450 km south of New Zealand’s main islands [31]. Thus, its occurrence on the remote Auckland Island might be a consequence of an ongoing southward dispersal.

In contrast, *P*. *zealandica* possesses several haplotypes, which are well-differentiated from each other by several mutational steps and exhibit a clear spatial structure. The genetic diversity of this species decreases from North Island to South Island, making an origin on North Island and a later dispersal to South Island the most probable scenario. *Pseudolycoriella zealandica* clearly manifests a pattern of northern richness and southern purity (also known as out-of-north pattern; cf. [18]). The tripartite spatial haplotype structure of *P*. *tewaipounamu* repeats this pattern, although the species is restricted to South Island. The phylogeographic structure within this species might be explained by the last ice age: two genetic lineages, estimated to be at least 112 ka old (node 42 in Figure 6), were forced to retreat into northern and south-eastern refugia on South Island as the ice shield of the Southern Alps reached its largest extent during the Last Glacial Maximum (LGM, Figure 2B). Shulmeister et al. [32] found evidence for a gradual warming during the early deglaciation instead of an abrupt warming. Thus, the retreat of the Southern Alps’ ice sheet presumably started earlier on its northern margin. This might have allowed a leading-edge southwards expansion out of the north-western refugium, while the south-eastern lineage was still trapped (Figure 2C). It should be mentioned in this context that the Catlins (i.e., the most south-eastern part of South Island) have an extraordinary inventory of *Pseudolycoriella* species. Besides the aforementioned *P*. *tewaipounamu*, five other *Pseudolycoriella* species were collected there. Three of them (*P*. *hauta*, *P*. *plicitegmenta*, and *P*. *porehu*) were found exclusively in this region and seem to be endemic [16].

In the case of *P*. *sudhausi*, a recolonisation of the areas formerly covered by glaciers occurred not only from the North but also from the South. During postglacial range expansion, the lineages originating in the south-western and the western parts of South Island spread with the advancing forest habitats, finally meeting at the Divide (a pass of 532 m asl; Figure 3B) after approaching through two valley systems from opposite sides of the pass. However, a subsequent fusion of these lineages was probably prevented by high-density blocking [33]. The high density of specimens of a single lineage within the respective populations reduces the probability that immigrants from other populations will find a suitable mate. Consequently, the possibility of them successfully reproducing with members of the local population is greatly reduced.

The intraspecific phylogeographic patterns of these *Pseudolycoriella* species resemble those known for other New Zealand taxa. Thus, a phylogeographic break along Cook Strait has been observed for several insect species, such as *Kikihia subalpina* (Cicadidae, Hemiptera) [34] and *Talitropsis sedilotti* (Rhaphidophoridae, Orthoptera) [35]. Examples of spatially structured subpopulations on South Island were found in *K*. *subalpina* [34], as well as the zopherid beetles *Epistranus lawsoni* and *Pristoderus bakewelli* (Zopheridae, Coleoptera) [36]. Consequently, if a sufficient degree of taxonomic knowledge exists, Sciaridae is also a group that is well-suited for biogeographic analyses. This is particularly due to their high (albeit passive) mobility and their ability to quickly establish dense and large populations, which apparently do not intermix with other intraspecific lineages at secondary contact zones [37].

### 4.2. Island Hopping and Speciation

Speciation events can only occur on one of the main islands due to the (almost) persistent separation of these islands and the limited gene flow between the separate *Pseudolycoriella* lineages (see above). In the case of a species distributed throughout New Zealand, the speciation must therefore have been followed by a colonisation event. Hence, two principal basic types of colonisation have to be distinguished within New Zealand: from North Island to South Island, and vice versa. In most cases, the direction of these colonisation events can be deduced from the region of origin of the respective lineage as revealed by RASP analyses, similar to the interpretation approach applied above to *P*. *tonnoiri*, where an intraspecific differentiation on North Island was followed by a southward dispersal. Similar reasoning can also be applied to *P*. *macrotegmenta* at the intraspecific level, where a southward dispersal most likely took place before differentiation into the two island-specific lineages (branch {14,15} in Figure 4).

Besides these biogeographically clear cases, RASP analyses did not always reveal a conclusive indication of the regions of origin of consecutive species. Thus, the results for the closely related species *P*. *gonotegmenta*, *P*. *plicitegmenta*, *P*. *robustotegmenta*, and *P*. *subtilitegmenta* (outgoing from node 10 in Figure 4 and Figure 6A) suggest a sequence of MRCAs with a distribution across entire New Zealand (node 11 and 12). The existence of descendants of a widespread species which also occur on both large islands would contradict our hypothesis of limited dispersal and thus a restricted gene flow between these islands. This scenario would require additional colonisation and extinction events and thus violate the principle of parsimony. However, the probability values for the origin of the root (node 10) of this four-species complex do not allow a clear distinction between the scenarios that the MRCA inhabited entire New Zealand (Figure 6B) or solely North Island (Figure 6C). Thus, a southward dispersal event might be assigned to branch {9,10} or branch {10,11}. The biogeographic events outgoing from node 11 also remain vague. Starting with the ancient species at node 11, which most likely existed across entire New Zealand, we have to account for at least one dispersal event (Figure 6D–E). This dispersal must have taken place either during the anagenesis of *P*. *gonotegmenta* (Figure 6D) or during the existence of its adelphotaxon prior to the differentiation at node 12 (Figure 6E). However, we could not deduce the direction, because it was not possible to assign the species split at node 11 to any region. A third possibility is a vicariance event which took place during the time of differentiation at node 11 (Figure 6F). In this case, the adelphotaxon of *P*. *gonotegmenta* (i.e., branch {11,12}) inhabited the southern part of South Island, from where it must have rapidly dispersed northwards. The dating estimate according to *Beast analysis supports such a scenario: an estimated mean age of 30.4 ka (95% HPD interval: 3.2–68.6 ka) is given for node 11. Thus, the species’ split coincides with the time frame of the LGM (29–31 ka according to Williams et al. [38]), when New Zealand’s islands were connected and the Southern Alps were largely covered by glaciers. A spatial separation of populations by these glaciers might be regarded as a plausible cause for the differentiation into two species initiated at node 11. Nevertheless, we consider this particular dispersal event to have an undetermined direction and that it occurred after the time frame of node 11. In total, the *P*. *macrotegmenta* s. str. clade—all species descend from node 5—has most likely undergone three North-to-South dispersal events and one of unknown direction.

Several dispersal events were also detectable for the second monophyletic group of autochthonous New Zealand *Pseudolycoriella* species. A dispersal event at the intraspecific level was already shown above for *P*. *zealandica*, assigned to branch {21,22}. A second intraspecific dispersal, with the same direction, was detected for *P*. *dagae*. Above the species level, dispersal events were identified during the existence of branches {18,19}, {24,32}, {26,27}, and {39,40} (Figure 5). The dispersal event during the existence of branch {39,40} is particularly important, because it is the only one that was unequivocally in the direction of South Island to North Island.

At the root of the *P*. *jejuna* clade, a similar situation exists to that described above for the *P*. *gonotegmenta* species group. RASP analysis again reveals an entire-New-Zealand distribution of the MRCA and also for one of its descendants (node 33 and 37 in Figure 5 and Figure 7). The MRCA was either distributed across entire New Zealand or solely on North Island. Thus, two scenarios are plausible: a southward dispersal before the split at node 33 (Figure 7A), or this dispersal event occurring in the time frame between split 33 and 37 (Figure 7B). Both scenarios require three dispersal events: in the first scenario, three south-directed events, and in the second scenario, one south-directed and two north-directed dispersal events. On average, the probability values of ancestral species ranges are higher in the first scenario. However, because of this ambiguity, we instead assign the first southward dispersal event on the composite branches {23,33,37} to a single branch. The subsequent events on branches {37,43,44} and {44,present} were regarded as events for which the direction of dispersal cannot be resolved.

Branches to which biogeographical events could be assigned were taken from both chronograms and are shown comparatively in Figure 8. A total of 13 colonisation events can be shown for both *Pseudolycoriella* species groups, with a clear directional disparity: nine events had a southern direction, one a northern direction, and three an unresolved direction. However, this picture is still incomplete, because genetic data are not available for all known species. Three of these genetically unsampled taxa inhabit North Island, and two South Island. Existing morphological data unfortunately do not allow their precise placement in the phylogenetic system (compare Figure 61 in Köhler [16]). Nevertheless, at least one further southward dispersal event has to be assumed for one species from South Island (i.e., *P*. *porehu*), because morphology implies a closer relationship with the northern *P*. *orite* than the southern *P*. *mahanga* [16].

The dispersal events within New Zealand do not show any clear temporal pattern (Figure 8). Since the initial colonisation of New Zealand (see below), dispersal across the sea straits separating the main islands (Kuripapango Strait during the late Miocene–early Pliocene; Manawatu Strait during the late Pliocene; Cook Strait in recent years [8]), seems to be evenly distributed from the late Miocene to the present. Presumably, the number of events is too low and their resolution in time is not fine enough for the identification of possible periods with increased colonisation rates.

Due to imbalance in the migration direction, North Island has to be regarded as a radiation centre, while South Island is a receptor of immigrating taxa. Thus, taxa invading South Island from the North rarely speciated and even more rarely recolonised North Island. Consequently, the pattern of northern richness vs. southern purity, already demonstrated at the intraspecific level, is repeated at the species level.

In general, a poleward impoverishment of genetic diversity has commonly been observed, as shown by Hewitt [39] for Europe and North America, and is explained by the Pleistocene climatic cycles. The main reasons in the northern hemisphere are the extirpation of northern populations and the concentration of populations in southern refugia during glacial periods, and genetic bottlenecks during leading-edge expansions during interglacials. These general effects also apply to New Zealand, of course, with the compass directions reversed. However, forest communities persisted even in the southern parts of South Island during glacial maxima [4], speaking against a complete extinction of all southern populations of *Pseudolycoriella*. This assumption of constant survival of *Pseudolycoriella* even in the extreme South of New Zealand is clearly supported by our genetic data and also by the existence of species such as *P*. *hauta*, *P*. *plicitegmenta*, and *P*. *porehu*, which occur exclusively in this region.

Hence, an increased extinction rate cannot be considered to be the main cause of the lower biodiversity on South Island. As an alternative, a reduced speciation rate has to be taken into consideration, as observed by Buckley et al. [6] for zopherid beetles during the Miocene and Pliocene for New Zealand as a whole. At first glance, the presence of multiple forest refugia should lead to a greater number of geographically separated populations and thus to a higher probability of speciation events, but also to genetic bottlenecks. Furthermore, all southern populations presumably faced a time-lag in their postglacial dispersal opportunities, as we suggested for *P*. *tewaipounamu*. Therefore, southern populations could not colonise such large areas as their northern counterparts during interglacials. Consequently, they were not able to disperse to as many areas that became refugia during subsequent ice ages, which resulted in an on-average lower chance of becoming separated and erecting reproductive barriers within their ranges.

Although the existence of undiscovered *Pseudolycoriella* species is not unlikely, we consider our conclusions to be plausible. For one thing, we do not expect many undiscovered species, because although the focus of the fieldwork was on South Island, this was not reflected in a higher number of species compared to North Island. Furthermore, we consider it highly unlikely that our sample was so biased that the discovery of additional species would lead to significantly different patterns.

### 4.3. Colonisation of New Zealand

The most likely areas of origin for the colonisation of New Zealand are Australia and New Caledonia. Phylogenetic connections between Australia and New Zealand have been frequently observed, and were, for example, demonstrated for the well-known bioluminescent fungus gnat genus *Arachnocampa* (Keroplatidae, Diptera) [40]. However, links to New Caledonia are also possible, as for example known in New Zealand cicadas, for which two independent colonisations were identified: one from Australia and one from New Caledonia [41]. Although six publications from the last decade address the occurrence of *Pseudolycoriella* species in the countries neighbouring New Zealand, reporting 17 species for Australia and 14 for New Caledonia [42,43,44,45,46,47], the sciarid fauna of the Australasian and wider Pacific region is still poorly understood. Furthermore, all these studies are exclusively based on morphological analyses, which currently do not allow species-level hypotheses of relationships to be formulated, as would be possible with the use of appropriate molecular markers. Therefore, it was not possible to identify phylogenetic relationships outside New Zealand. The inclusion of three undescribed species from Tasmania in our study revealed that these represent a monophyletic group within the *P*. *macrotegmenta* lineage comprising New Zealand species. Thus, we must ask how often New Zealand was colonised by this genus.

In light of the fact that the genus *Pseudolycoriella* did not originate in New Zealand, as shown by the phylogenetic tree of Köhler [16], two mutually exclusive scenarios arise: two colonisation events by the MRCAs of the two monophyletic groups with a subsequent (re)colonisation of Tasmania starting from New Zealand, or three colonisation events. In the latter case, the MRCA of the complete *P*. *macrotegmenta* lineage would also have originated in Tasmania/Eastern Australia.

To assess the likelihood of each possibility, we have to consider the ancient meteorological, oceanographic, and climatic conditions in the New Zealand region. An eastward aeolian or pleustonic transport of tiny dipterans from eastern Australia to New Zealand is easily conceivable because of (i) the steady westerly wind regime caused by the mid-latitude westerlies [48] as well as (ii) the eastward sea currents (eastern extension of the East Auckland Current between Australia and the northern tip of New Zealand [49] and the Subtropical Front reaching the southern parts of New Zealand [50]). Furthermore, these eastward-directed meteorological and oceanographic constellations are not of recent origin. Thus, the wind regime is assumed to already have existed in the Pliocene, as Li et al. [51] revealed that the westerlies experienced a poleward shift of 1.9 degrees 3.3 to 3.0 Ma ago. The sea currents reaching New Zealand are extensions of major global currents such as the South Pacific Gyre or the Antarctic Circumpolar Current, which also existed several million years ago [52].

Assuming that the probability of a single dispersal event is not homogeneously distributed over the associated branch of the chronogram, but rather that the colonisation events precede the splitting of the species only by a short time span, the first colonisation of New Zealand by an ancient *Pseudolycoriella* species might be hypothesised to have occurred approximately 10 Ma ago (on branch {past,18}). This is later than the mid-Miocene climatic optimum (~16.9–14.7 Ma [53]), after which the global climate started cooling down [54]. This cooling has also led to changes in New Zealand’s vegetation structure [4,55,56], for example towards the dominance of southern beeches (*Nothofagus* spp.) in forest communities [55]. During this period, the rainforests in inland south-eastern Australia decreased, but the rainforest communities on the east coast persisted [57], which supports the hypothesis that the latter have been highly suitable for gnats and therefore might have served as a donor region for individuals drifting to New Zealand. For the other two potential colonisation events dated during the late Pliocene or to the Pliocene–Pleistocene boundary (on branches {past,1} and {1,2}), further cooling must be taken into consideration. Prebble et al. [56] dated the second major cooling episode of the last 30 Ma to this time frame. That the events in question are likely to have occurred during cooling periods, when species’ ranges shift towards the equator [39], also points to an origin in temperate regions of Australia and not in tropical New Caledonia. However, the presumed steady patterns of air and water circulation cannot automatically be considered to have led to successful colonisation events, because these can be prevented by high-density blocking, as shown at a more regional and intraspecific level in the case of *P*. *sudhausi*. Consequently, in order to successfully colonise, arriving individuals would have to have a competitive advantage over the indigenous species in utilising the available resources. If climatic changes did not occur simultaneously in New Zealand and south-eastern Australia, the ancestors of today’s New Zealand *Pseudolycoriella* species may already have been better adapted to cooler climates than New Zealand species using the same resources.

On the basis of all these considerations, we think that it is more likely that New Zealand was colonised three times by ancient *Pseudolycoriella* species originating from Australia. In this respect, our gnats are not special and join numerous examples of New Zealand insect taxa that originated in Australia [5]. Nevertheless, further faunistic and taxonomic work is needed on the sciarids of the Australian and the Oceanian realms, thus allowing increased use of Sciaridae to address biogeographic assessments.

## Figures and Tables

**Figure 1 insects-14-00548-f001:**
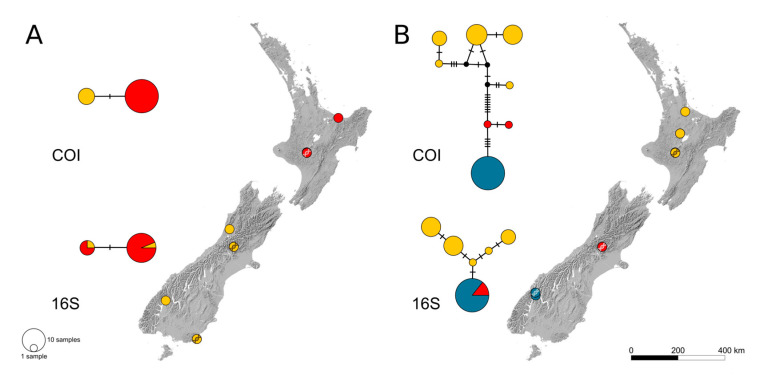
Haplotype networks (top COI, bottom 16S) and sample locations of (**A**) *Pseudolycoriella tonnoiri* and (**B**) *Pseudolycoriella zealandica*. Map sourced from the LINZ Data Service (CC BY 4.0 licence). Colours indicate different populations. Localities with successfully barcoded specimens are indicated by a DNA symbol.

**Figure 2 insects-14-00548-f002:**
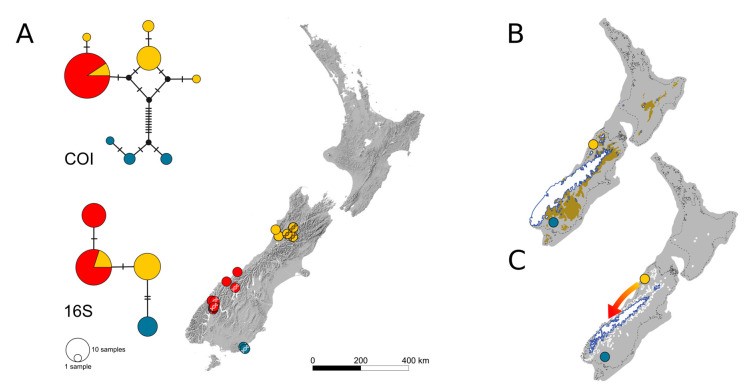
(**A**) Haplotype networks (top COI, bottom 16S) and sample locations of *Pseudolycoriella tewaipounamu*. Colours indicate different populations. Localities with successfully barcoded specimens are indicated by a DNA symbol. Map sourced from the LINZ Data Service (CC BY 4.0 licence). (**B**) Hypothetically separated populations during the Last Glacial Maximum (LGM) after McGlone et al. [29]. Glaciers coloured in white, tundra in light brown. (**C**) Hypothetical glacier retreat after the LGM and potential dispersal (indicated by an arrow). Glacier extent after the −4 °C model of Golledge et al. [30]. The ice shield extent during LGM is outlined in white.

**Figure 3 insects-14-00548-f003:**
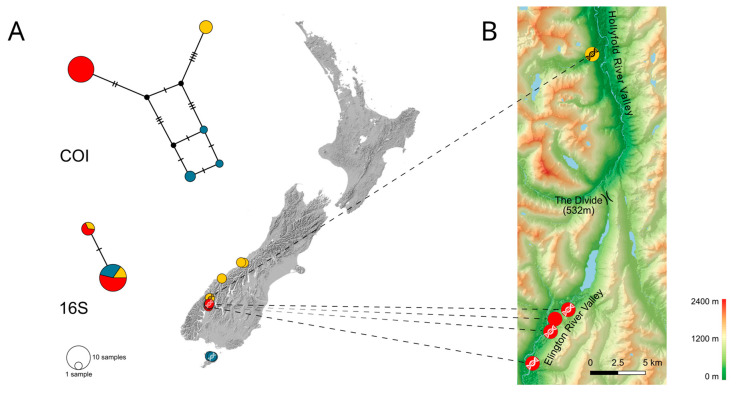
(**A**) Haplotype networks (top COI, bottom 16S) and sample locations of *Pseudolycoriella sudhausi*. Colours indicate different populations. Localities with successfully barcoded specimens are indicated by a DNA symbol. Map sourced from the LINZ Data Service (CC BY 4.0 licence). (**B**) Elevation map of the contact zone of two lineages of *P*. *sudhausi* in the south-east of South Island. Map based on ASTER GDEM data, provided by the Ministry of Economy Trade and Industry, Japan, and the National Aeronautics and Space Administration, USA.

**Figure 4 insects-14-00548-f004:**
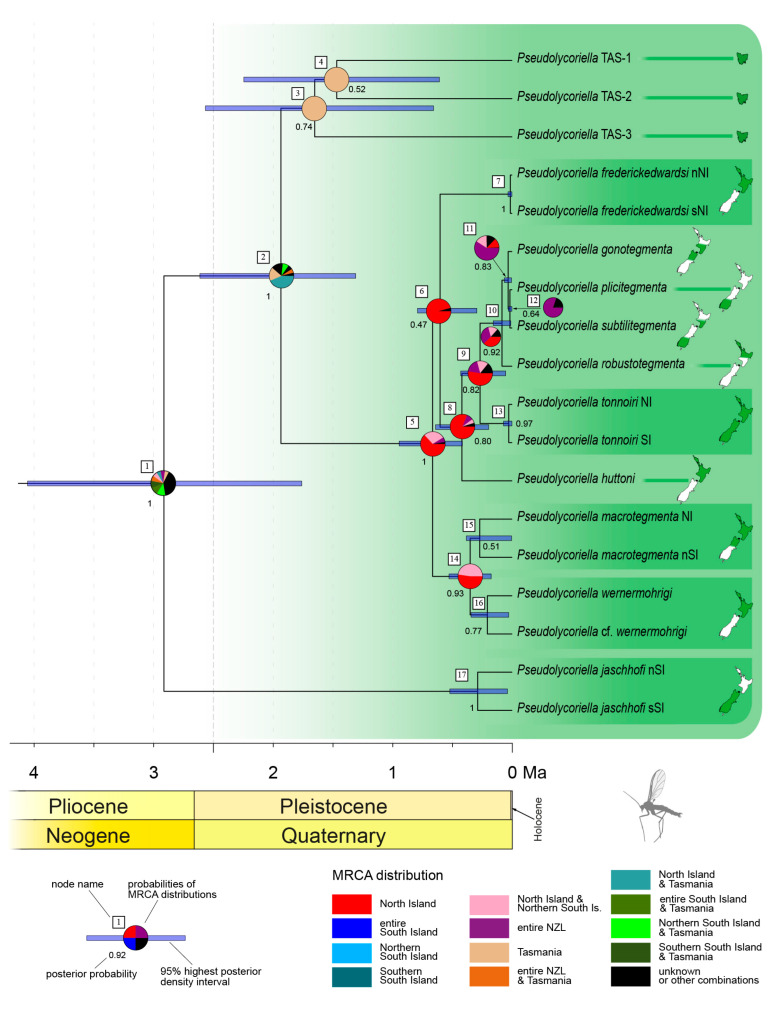
Chronogram of the *Pseudolycoriella macrotegmenta* species clade according to a *BEAST analysis (based on COI, 16S, and 28S). For each interspecific node, the probability of distributions of the respective taxon is depicted as a pie chart obtained from a RASP analysis (S-DEC). The distribution of each extant species is given by inserted maps (NI, North Island; nNI, northern North Island; nSI, northern South Island; SI, South Island; sNI, southern North Island; TAS, Tasmania).

**Figure 5 insects-14-00548-f005:**
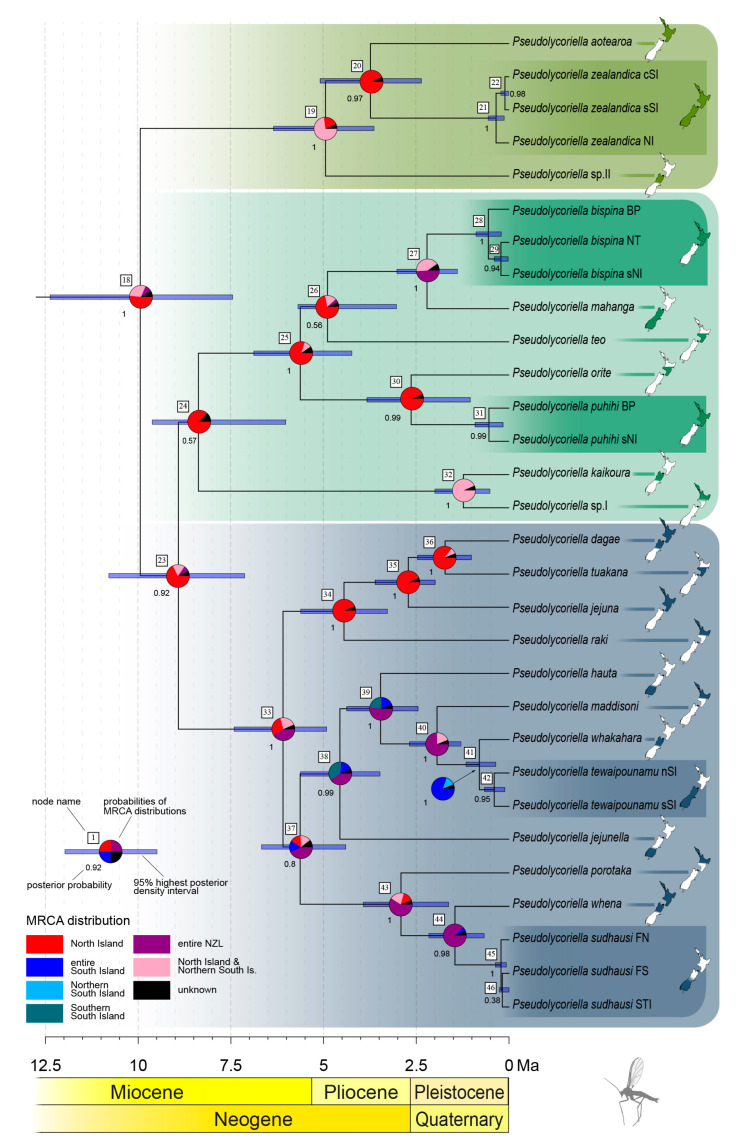
Chronogram consisting of the *Pseudolycoriella zealandica*, *P*. *bispina*, and *P*. *jejuna* clades according to a *BEAST analysis (based on COI; 16S, and 28S). For each interspecific node, the probability of the stem species’ distribution is depicted as a pie chart obtained from a RASP analysis (S-DEC). The distribution of each extant species is given by inserted maps (BP, Bay of Plenty; cSI, central South Island; FN, Fiordland North; FS, Fiordland South; NI, North Island; nSI, northern South Island; SI, South Island; NT, Taupo North; sNI, southern North Island; sSI, southern South Island; STI, Stewart Island).

**Figure 6 insects-14-00548-f006:**
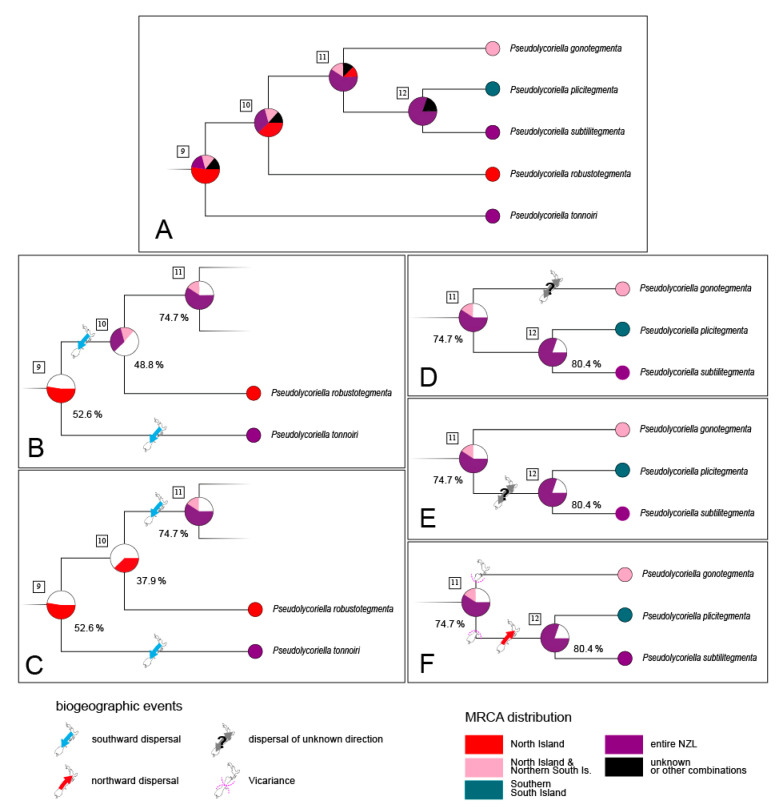
(**A**) Detail of Figure 4. (**B**,**C**) Possible scenarios for the current distribution patterns outgoing from node 9. (**D**–**F**) Possible scenarios for the current distribution patterns outgoing from node 11. Used colour codes for biogeographic events and (stem) species distributions are given below. At each node, the probability values for the considered MRCA distribution are given.

**Figure 7 insects-14-00548-f007:**
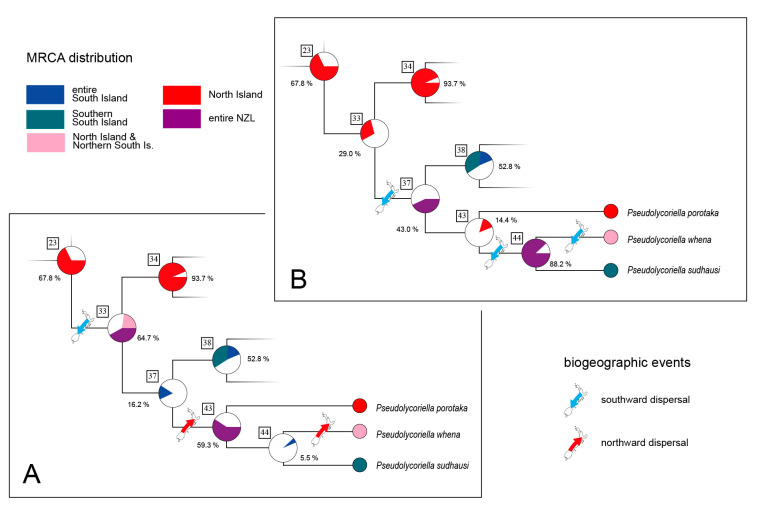
(**A**,**B**) Two possible scenarios for the current distribution patterns outgoing from node 23 in Figure 6. Used colour codes for biogeographic events and MRCA distributions are given. At each node, the probability values for the considered MRCA distribution are given.

**Figure 8 insects-14-00548-f008:**
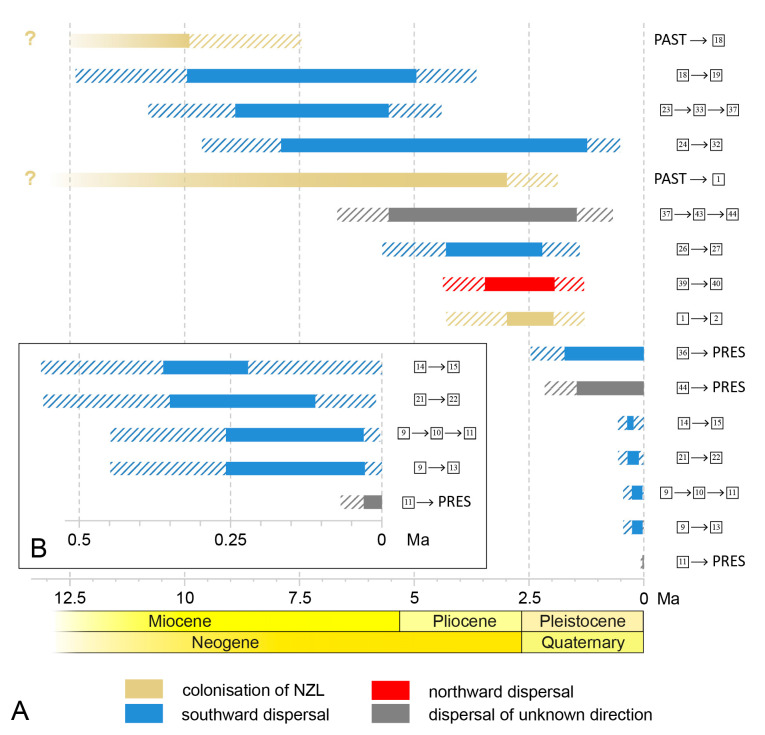
(**A**) Temporal comparison of all branches with dated biogeographic events from the chronograms in Figure 4 and Figure 6. (**B**) Enlargement of the five most recent events. The branches are named after their bordering nodes given on the right. The outer 95% HPD interval of each branch bounding node is hatched. Used colour codes for biogeographic events are given below.

**Table 1 insects-14-00548-t001:** Distribution of 26 out of 38 *Pseudolycoriella* species found at more than one sample locality across New Zealand according to Köhler [16]. A probably introduced species is marked with an asterisk.

Widely Distributed on Both Islands	Distributed on North Island and Northern South Island	Endemic to North Island	Endemic to South Island
*P. cavatica* *	*P*. *dagae*	*P*. *aoteraoa*	*P*. *fiordlandia*
*P*. *subtilitegmenta*	*P*. *gonotegmenta*	*P*. *bispina*	*P*. *jaschhofi*
*P*. *tonnoiri*	*P*. *macrotegmenta*	*P*. *breviseta*	*P*. *jejunella*
*P*. *zealandica*	*P*. *whena*	*P*. *frederickedwardsi*	*P*. *mahanga*
		*P*. *jejuna*	*P*. *sudhausi*
		*P*. *maddisoni*	*P*. *tewaipounamu*
		*P*. *orite*	
		*P*. *porotaka*	
		*P*. *puhihi*	
		*P*. *raki*	
		*P*. *robustotegmenta*	
		*P*. *wernermohrigi*	

## Data Availability

The genetic data presented in this study are available on GenBank (accession numbers are given in Appendix A) and at [58].

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
