# Peer review of "Northern Richness, Southern Dead End—Origin and Dispersal Events of Pseudolycoriella (Sciaridae, Diptera) between New Zealand’s Main Islands"

_insects, 2023, doi:10.3390/insects14060548_

Round 1
Reviewer 1 Report
The study is very good and will be of considerable interest to the small community of sciarid workers. I have only a minor comment and I don't feel I have enough expertise in phylogeographic analysis to comment specifically on those aspects (Editor I suggest you engage a reviewer who has this expertise).
Have the authors considered how many Pseudolycoriella species remain to be collected and described from New Zealand? The study is based on collections by the Jaschhof's and Maddison and there seems little doubt that there are more species awaiting discovery and description. Perhaps the collections in some localities were more thorough than in others and only during certain months of the year. I think it would be useful (in the Discussion) to state if the conclusions drawn from the analysis may be considered reliable/solid, if the analysis is based on many (but very likely not all) NZ Pseudolycoriella species?
Author Response
Dear reviewer,
Thanks a lot for your valuable comments! To meet your correct comment, we have added the following passage at the end of section 4.2 (Island-hopping and speciation):
“Although the existence of undiscovered Pseudolycoriella species is not unlikely, we consider our conclusions to be plausible. For one thing, we do not expect many undiscovered species, because although the focus of the fieldwork was on South Island, this was not reflected in a higher number of species compared to North Island. Furthermore, we consider it highly unlikely that our sample was so biased that discovery of additional species would lead to significantly different patterns.”
All the Best
Arne Köhler
The study is very good and will be of considerable interest to the small community of sciarid workers. I have only a minor comment and I don't feel I have enough expertise in phylogeographic analysis to comment specifically on those aspects (Editor I suggest you engage a reviewer who has this expertise).
Have the authors considered how many Pseudolycoriella species remain to be collected and described from New Zealand? The study is based on collections by the Jaschhof's and Maddison and there seems little doubt that there are more species awaiting discovery and description. Perhaps the collections in some localities were more thorough than in others and only during certain months of the year. I think it would be useful (in the Discussion) to state if the conclusions drawn from the analysis may be considered reliable/solid, if the analysis is based on many (but very likely not all) NZ Pseudolycoriella species?
We added the passage: at the end of section 4.2 (Island-hopping and speciation):
“Although the existence of undiscovered Pseudolycoriella species is not unlikely, we consider our conclusions to be plausible. For one thing, we do not expect many undiscovered species, because although the focus of the fieldwork was on South Island, this was not reflected in a higher number of species compared to North Island. Furthermore, we consider it highly unlikely that our sample was so biased that discovery of additional species would lead to significantly different patterns.”
Reviewer 2 Report
To reveal the biogeographic history of the New Zealand members of the sciarid genus Pseudolycoriella, the authors made a study on the described and undescribed species of Ps. According to the locations of each species, they divided congeneric species into four major biogeographic patterns, and then they made the haplotype and phylogeographic analyses by using three molecular markers. They identified 13 dispersal events in the NZ’s main islands, in which nine dispersal events were south-directed, thus the North Island was considered to be the centre of radiation for this genus. They also made an estimate on the evolutionary divergence time of this genus. On the whole, this is an interesting study, which firstly took the sciarid genus Pseudolycoriella for the study of species dispersal between North and South Islands. The research results are suitable for publication in this journal, but the MS still has some problems in the statements. They are listed as follows:
1. Lines 19-20
“To reveal the biogeographic history of the New Zealand members 19 of the sciarid genus Pseudolycoriella, we analysed three molecular markers in a Bayesian approach.” This statement should be clarified more clearly.
2. Lines 30-31
There are eight key words, it’s better to list no more than five.
3. Lines 78-81
“Of the 37 autochthonous and endemic New Zealand species (both described and undescribed), 25 are known from more than one sample site. These species can be roughly assigned to four major biogeographic patterns (Table 1)”.
Firstly, the biogeographic patterns of Pseudolycoriella cannot be taken as the introduction part, which should be moved to the method section.
Secondly, you divided these species into four biogeographic patterns. What’s the theory for this division?
4. Lines 110-113
Why do you select only three genes (COI, 16S and 28S) for the analyses of colonization, population dispersal? You’d better clarify the reason why these genes are suitable for this research.
5. Line 118
What are the version and parameters of PopART employed in your analysis?
6. Lines 166-168
Figure 1
What does the colored circle indicates? What does the symbol in the circle means? The legend should be illustrated in more detail.
The similar issues are also present in the legends of “Figure 2 and 3”.
7. Lines 263-264
This sentence is difficult to understand. Please make it clear.
8. Line 266
This statement is better to be “In P. tonnoiri the lineages differ in only a single base substitution”.
9. Lines 303-306, 335-339
These sentences are too complicated to understand, which should be simplified or divided into several short sentences.
Author Response
Dear reviewer,
Thanks a lot for your valuable comments! We have implemented all your comments. The only point we would like to object to is to move Table 1 and the corresponding explanatory text to the methods section. The table and the short text section merely summarise the paper by Köhler (2019). When writing the first draft of the manuscript, we have also debated whether this summary belongs to the results section and discussion section or not. However, as this summary is actually only based on one paper (and thus not a real review) and shows the inhomogeneous distribution of gnat species in New Zealand, it directly points to our research question. Therefore, we would like to leave this passage in the introduction
We also agree with your comment that it is premature to speak of “biogeograhic patterns” in this context. Instead, we change to the more neutral term "distribution".
All the Best
Arne Köhler
To reveal the biogeographic history of the New Zealand members of the sciarid genus Pseudolycoriella, the authors made a study on the described and undescribed species of Ps. According to the locations of each species, they divided congeneric species into four major biogeographic patterns, and then they made the haplotype and phylogeographic analyses by using three molecular markers. They identified 13 dispersal events in the NZ’s main islands, in which nine dispersal events were south-directed, thus the North Island was considered to be the centre of radiation for this genus. They also made an estimate on the evolutionary divergence time of this genus. On the whole, this is an interesting study, which firstly took the sciarid genus Pseudolycoriella for the study of species dispersal between North and South Islands. The research results are suitable for publication in this journal, but the MS still has some problems in the statements. They are listed as follows:
- Lines 19-20
“To reveal the biogeographic history of the New Zealand members 19 of the sciarid genus Pseudolycoriella, we analysed three molecular markers in a Bayesian approach.” This statement should be clarified more clearly.
We change the sentence to: “To reveal the biogeographic history of the New Zealand members of the sciarid genus Pseudolycoriella, we analysed three molecular markers of selected species and populations in a Bayesian approach.”
- Lines 30-31
There are eight key words, it’s better to list no more than five.
We delete three of them, i.e. *Beast; biogeography; Sciaroidea
- Lines 78-81
“Of the 37 autochthonous and endemic New Zealand species (both described and undescribed), 25 are known from more than one sample site. These species can be roughly assigned to four major biogeographic patterns (Table 1)”.
Firstly, the biogeographic patterns of Pseudolycoriella cannot be taken as the introduction part, which should be moved to the method section.
Secondly, you divided these species into four biogeographic patterns. What’s the theory for this division?
We would like to object to move Table 1 and the corresponding explanatory text to the methods section. The table and the short text section merely summarise the paper by Köhler (2019). When writing the first draft of the manuscript, we have also debated whether this summary belongs to the results section and discussion section or not. However, as this summary is actually only based on one paper (and thus not a real review) and shows the inhomogeneous distribution of gnat species in New Zealand, it directly points to our research question. Therefore, we would like to leave this passage in the introduction.
We also agree with your comment that it is premature to speak of “biogeograhic patterns” in this context. Instead, we change to the more neutral term "distribution".
- Lines 110-113
Why do you select only three genes (COI, 16S and 28S) for the analyses of colonization, population dispersal? You’d better clarify the reason why these genes are suitable for this research.
New Text: “The gene selection is based on the first phylogenetic study of sciarids by Shin et al [11]. This mixture of rapidly evolving mitochondrial genes and a more conservative nuclear gene seems to be appropriate for the time scale under investigation, as already applied by Köhler [16]. In some cases, residual DNA from the DNA-extraction by Köhler [16] was re-sequenced to eliminate ambiguities in COI or 16S sequence of some specimens.”
- Line 118
What are the version and parameters of PopART employed in your analysis?
“To enable intraspecific spatial analyses, the software Popart 1.7 [20] was used to generate median-joining haplotype networks [21] based on COI and 16S sequences.”
New reference:
- Bandelt, H.-J., Forster, P.; Röhl, A. Median-joining networks for inferring intraspecific phylogenies. Molecular Biology and Evolution 1999, 16, 37–48, https://doi.org/10.1093/oxfordjournals.molbev.a026036
- Lines 166-168
Figure 1
What does the colored circle indicates? What does the symbol in the circle means? The legend should be illustrated in more detail.
The similar issues are also present in the legends of “Figure 2 and 3”.
Changed caption:
“Figure 1. Haplotype networks (top COI, bottom 16S) and sample locations of (A) Pseudolycoriella tonnoiri and (B) Pseudolycoriella zealandica. Map sourced from the LINZ Data Service (CC BY 4.0 licence). Colours indicate different populations. Localities with successfully barcoded specimens are indicated by a DNA symbol.”
Figs 2 and 3 changed accordingly
- Lines 263-264
This sentence is difficult to understand. Please make it clear.
New: “The two analysed species which are distributed across both large islands, i.e. P. tonnoiri and P. zealandica, both show spatial structuring of their haplotypes, with a clear split along Cook Strait.”
- Line 266
This statement is better to be “In P. tonnoiri the lineages differ in only a single base substitution”.
Done.
- Lines 303-306, 335-339
These sentences are too complicated to understand, which should be simplified or divided into several short sentences.
New: “The high density of specimens of a single lineage within the respective populations reduces the probability that immigrants from other populations will find a suitable mate. Consequently, the possibility of them successfully reproducing with members of the local population is greatly reduced.”
New: “The existence of descendants of a widespread species which also occur on both large islands would contradict our hypothesis of limited dispersal and thus a restricted gene flow between these islands. This scenario would require additional colonisation and extinction events and thus violate the principle of parsimony.”